# Eukaryotic CRFK Cells Motion Characterized with Atomic Force Microscopy

**DOI:** 10.3390/ijms232214369

**Published:** 2022-11-19

**Authors:** María Zamora-Ceballos, Juan Bárcena, Johann Mertens

**Affiliations:** 1Centro de Investigación en Sanidad Animal (CISA-INIA/CSIC), Valdeolmos, 28130 Madrid, Spain; 2Madrid Institute for Advanced Studies in Nanoscience (IMDEA Nanoscience), Campus de Cantoblanco, 28049 Madrid, Spain

**Keywords:** atomic force microscopy, eukaryotic cells, migration, motion, mechanical properties, spring constant, cytoskeleton, topography

## Abstract

We performed a time-lapse imaging with atomic force microscopy (AFM) of the motion of eukaryotic CRFK (Crandell-Rees Feline Kidney) cells adhered onto a glass surface and anchored to other cells in culture medium at 37 °C. The main finding is a gradient in the spring constant of the actomyosin cortex along the cells axis. The rigidity increases at the rear of the cells during motion. This observation as well as a dramatic decrease of the volume suggests that cells may organize a dissymmetry in the skeleton network to expulse water and drive actively the rear edge.

## 1. Introduction

Cell migration is a necessary function in organisms that contributes to several important physiological processes, from embryogenesis to immune response development, but also a wide variety of pathologies [1]. Without the ability to move, cells would not grow, divide, or be recruited. Thus, understanding the molecular and biophysical mechanisms underlying cell migration is fundamental for disease treatment, cellular transplantation, and tissue engineering. Over the past few years, outstanding progress have been made in understanding the complexity of actin assembly regulation at the leading edge of moving cells, and microtubule polymerization [2]. Cell migration has been extensively studied on 2D surfaces because this reduces the complexity of cellular processes visualization and calculation of traction or detachment forces applied to the substrate.

To be able to move, first, a cell must be attached to a surface. Then, it grows some protrusions forward to exert the traction force it needs. This process polarizes spatially the cell between the front (closest to the direction of migration, or leading edge) and the rear (the opposite to the front, or tail). While the front is crawling, the rear must detach from the surface, allowing it to follow motion forward. Therefore, cell locomotion is a process that converts chemical energy into mechanical force [3,4].

Cell motility is foremost a mechanical phenomenon characterized by a balance between counteracting traction and adhesion forces. Movement is originated from changes in mechanical state of the cytoplasm (cytoskeleton), such as gel/sol transformations, and results in a coordinated cycle of protrusion, attachment, and retraction [5]. Leading edge protuberances are normally generated by controlled actin networks assembly, while adhesion and retraction rely on tension generated by actin–myosin interactions. Characterization of the forces exerted by migrating cells and their mechanical properties complement biochemical and structural data. This level of characterization requires technologies that can recognize, quantify, and characterize dynamics of localized events that are below light microscopy resolution. Several techniques such as optical tweezers, micropipette aspiration, or cytometry have been adapted in order to probe these changes, but they are generally limited by spatial resolution [6].

Eukaryotic cells have developed a variety of migration modes relying on diverse traction-force-generation mechanisms. It generally involves drastic changes in cells shape which are driven by the cytoskeleton. This rely on global or local deformations of the cells surface. Morphological-related parameters such as height and volume variation are therefore basic physical properties that play a key role in cell functions and regulation [7,8]. However, it has been poorly investigated in cell biology so far, mostly because it is difficult to measure it precisely.

AFM represents a powerful tool to investigate cell mechanics since it allows for simultaneous measurements of local mechanical properties and topography of the living cell in aqueous environment. Previous AFM studies were aimed to obtain high resolution topographic images of different cellular domains, to elucidate mechanical contributions of different cytoskeletal structures, and also to draw functional correlation between the state of the cell and its mechanical properties [9,10,11,12,13]. Ultimately, it has also been demonstrated the ability of AFM to map and characterize organelles and microorganisms inside cells, at the nanoscale [14,15]. Another AFM exciting possibility is that dynamic processes can be observed in real time [16].

In this article, we characterized mechanical properties of CRFK eukaryotic cells during their displacement by analyzing successive AFM images of the same cell. Furthermore, we reconstructed a real time movie of the process. We found that migrating cells shows a spatial dissymmetry in spring constant that polarize the cells along their motion axis. We suggest that this rigidity gradient onto cells surface leads to a dramatic change in cell morphology characterized by large volume decrease when the cytoskeleton contracts and expulse water. This contraction produces an active motion of the rear part of the cells in the direction of the front edge.

## 2. Results and Discussion

### 2.1. Topography

Time-lapse AFM images were recorded in cell culture medium (DMEM) at 37 °C. A total of 40 successive images were collected at fixed time intervals during 3 h 47 min. We monitored the topography and spring constant response of the cell during its motion at the glass slide surface.

Figure 1 shows the time-lapse imaging of a CRFK cell during its motion (See also Appendix A). As shown in the images of the Figure 1a, the cell topography changes while moving (mainly from left to right) at the surface of the glass. The cell adheres to and spreads on the glass substrate, which gives it a flat shape. Its average dimensions are 40 µm long and 3–4 µm high. Besides the cell position and shape, we also observed the presence of a large amount of membrane residue with 250 nm in height attached onto the glass surface at the rear of the cell (Figure 1a, red arrow, t = 6 min) that gradually disappears during the experiment. We also notice the presence of a protuberance (Figure 1a, green arrow, t = 6 min) at the right of the cell with ~600 nm in height and more than 20 µm in length that changes aspect with time, indicating a rotation of the cell during its motion. This rotation of the cell matches with the presence of another cell that blockades the initial trajectory. A new protuberance appears on the left of the initial one and leads the cell to its new direction.

### 2.2. Motion and Speed

In addition to the topography, we analyzed the nature of the cell motion during the experiment. The Figure 1b is a kymograph of the 40 height images representing the spatial position over time of the cell top during the experiment. The X-axis also represents time. It clearly shows the motion of the cell is about 45° from the X-axis (white arrow) during the first 90 min, then the change in cell’s trajectory along X-axis (red arrow). Color scale (Z-scale) shows the change in cell maximum height during its motion. Optical microscopy observation under the same conditions showed similar sizes and shapes and confirmed our observation of the nature of the cell motion measured with AFM (Figure 1c).

The position of the cell top was extracted from each image (See Section 3) to follow the trajectory of the cell from its initial position (Figure 2a). During the first 60 min (Figure 2a, frames 1 to 10), the cell moves linearly ~45° from X-axis. It travels 18 µm (Figure 2a, inset, red curve) with an average speed of ~310 nm/min (Figure 2a, inset, black curve). The second phase from 60 min < t < 130 min (Figure 2a, frames 11 to 24) is more complex as the cell changes its direction. This behavior was also observed by optical microscopy when cells do not have space to move forward because of high confluence. Here the cell is still moving, but it also rotates clockwise to orientate along a new protuberance following the X-axis. From t = 75 min, the motion is more stepped. The traveled distance is 21 µm at an average speed of 284 nm/min. While rotating, the cell changes its morphology, spreading and taking a more rounded shape between the old motion axis at the rear and the new one at the front (Figure 1a, t = 68 min).

Finally, from t = 130 min (Figure 2a, frames 25 to 40), the cell nearly stops. Its velocity decreases dramatically to 40 nm/min. This behavior corresponds to a phase where the cell trajectory appears more aleatory and shows random motility along Y-axis. The rear part of the cell stretches while it travels a 4 µm distance, mostly around a fixed position, until the end of the experiment at t = 228 min. This weakly correlated motion shows that the cell is destabilized by its change in direction and the creation of a new protuberance that affected the persistence of the motion. In total, the cell traveled 43 µm distance.

The speed of cell migration is mediated by reversible reactions among cell membrane adhesion receptors (e.g., integrins-ligands interaction) and dynamic of pseudopods. Fastest cells move up to 15 µm/min [17] but their speed covers a range as wide as their type and conformation. For example, fibroblasts migration in culture medium is slower with an average speed less than 1 μm/min. But most of the eukaryotic cells strongly adhere to glass surface and their speed is generally to the order of 10–1000 nm/min [18]. Optical microscope observation confirmed our measurements of speed (30–440 nm/min) and shape for CRFK cells in motion under the same configuration of adhesion and anchoring (Appendix A), but we notice that CRFK cells can also move up to 2000 nm/min depending on the cell spread area available, using amoebic mode migration. Besides conformation, we also observed the junctions that provide adhesion between neighboring cells. These structures form bridges of different morphologies and are known to ensure the exchange of ions and proteins through a regulated gate between cells [7]. They also are an important part in movement as the direction of motion is closely related to the shape and the position of the cells as well as their respective junction. In that sense, cells junctions play the role of the protuberance created by isolated cells in case of free motion. Cells confluence is therefore a key parameter in the characterization of cell migration. At large confluence, when cells are attached to each other, migration critically depends on cell–cell interactions coupled to a dynamic actin cytoskeleton. By extension, cell junction leads to collective cell migration, where a cohesive cells group coordinates cytoskeleton activity with the surrounding tissue.

### 2.3. Height and Volume

The faster motion phase occurs during the first 60 min and induces an important cell height decrease, from 3.8 µm to 2.6 µm (Figure 2b). Cell volume decreases dramatically about 45% at the same time (Figure 2b, inset). By extension, we may assume that cell volume is decreasing up to 55% at t = 90 min, during the phase where cell displacement is faster and more linear. Thereafter, cell height increases abruptly at t = 120 min from 2.6 µm to 3.2 µm at t = 130 min, while it slows down dramatically and nearly stops. Finally, cell height tends to increase again slightly up to 3.4 µm during the last 100 min of the experiment where the cell stays unmoved. We also observed some jumps in cell height, mainly at t = 60 and 110 min (Figure 2b, red arrows), which corresponds to a sudden change in cell front edge shape, characterized by a stretching and spreading during cell rotation.

Height changes over the course of the cell life cycle, but also on a much more rapid timescale in response to various perturbations such as external osmotic pressure, substrate stiffness, and cell spread area. Moreover, cell migration mechanisms rely on total or partial deformations of the cell and its volume is continuously affected by mechanical and morphological alterations. During our experiment, the external osmotic pressure and substrate stiffness are considered unchanged, while cell–substrate area decreases during the initial phase of motion (Figure 2b, inset, red curve). Therefore, cell height and volume variation should originate from another activator that do not rely on external stimuli. Mechanisms involved in cell size regulation include: (i) cell membrane proteins forming ionic channels and pumps, as well as (ii) cytoskeletal proteins for stiffness control. It causes water flow across the plasma membrane and the cell undergoes a process of swelling followed by regulatory volume decrease (RVD), or shrinking followed by regulatory volume increase (RVI). This mechanism regulates many cell functions, including migration, thus cell mechanical properties should be affected by volume regulation during migration [19].

### 2.4. Spring Constant

To get more insight in the process that generate motion and size changes, we performed more detailed study of the cell spring constant during its displacement. The cytoskeleton, which supports the membrane, is the component of the cell that makes cell movement possible. This network of fibers is spread throughout the cell’s cytoplasm and move cells from one location to another in a fashion that resembles crawling. The initial response of a cell to a migration-promoting agent is to polarize and extend protrusions in the direction of motion. Polarity refers to the front-rear polarity that represents the molecular and functional differences between the leading edge and the back of the cell. These protrusion adhesions serve as traction sites for migration as the cell moves forward over them, and they are disassembled at the cell rear, allowing it to detach.

The key parameters in cell motion process are therefore those that generate spatial polarity in the cell. We identified a symmetry breaker of our system by recording images of the spring constant k of the cell during the experiment (Figure 3a). Here we measured the deformation of the cell when the AFM tip is indenting it. In Eukaryotic cell, plasma membrane and cortex thicknesses are of the order of 10 nm and 150 nm respectively [20]. In our experiments, we applied forces up to 1.3 nN, indenting ~300 nm through the cell (Figure 3b, inset). Therefore, k-response corresponds mostly to the actomyosin cytoskeleton contraction, which is a key physical property of cell mechanics. Spring constant (k) profiles of the images along cell axis permit to analyze spatial and temporal evolution of k during the experiment (Figure 3b). At the start of the experiment, t = 0 min, rigidity mapping of the cell shows large heterogeneity between k_front_ (left) and k_rear_ (right) edge of the cell. At t = 0 min, the cell shows a k-dissymmetry with a linear increase in spring constant from k_rear_~0.2 nN/µm to about k_front_~0.8 nN/µm (Figure 3b, black line). At t = 68 min, k_rear_ has increased up to k_middle_~0.6 nN/µm (Figure 3b, red line). At t = 125 min, k is constant over the whole cell axis (Figure 3b, green line). At t = 222 min k has decreased almost at the center of the cell and dissymmetry pattern is newly present (Figure 3b, blue line).

For more clarity, we plotted the temporal evolution of spring constant for three points of the cell axis at the rear, middle, and the front edge of the cell axis (See methods and Figure 3c). As observed with profiles, the value of k_rear_ = 0.23 nN/µm (Figure 3c, blue line) is the lowest at t = 0 min, characterizing the k-dissymmetry along the cell with the value of k_front_ = 0.77 nN/µm (Figure 3c, red line). The first phase of motion (t < 60 min), where the cell moves linearly and its height decreases to 2.4 μm, is characterized by an increase in k_rear_ to 0.59 nN/µm. The second phase shows similar behavior for k_rear_ and k_middle_ (Figure 3c, green line), both increasing to 0.64 and 0.69 nN/µm respectively at t = 90 min. Thereafter, k tends to decrease for the whole cell, with k_front_ reaches 0.75 nN/µm at t = 130 min. As a consequence, there is no more dissymmetry in k between rear and front edge (Figure 3c, inset) when the cell almost stops at t = 130 min.

These results highlight the importance of the cell k gradient and its temporal and spatial evolution. The capability of cells to generate contractile forces originates from the activity of the molecular motor myosin II on its substrate, actin filaments [21,22]. Whereas actomyosin-mediated contractility in striated muscle is well understood, the modes of regulation and force transmission in cells are less certain [23]. Our finding also provides an example of how diverse mechanical properties at cellular scales can arise from locally activated common molecular components. In our experiment, the motion of the cell is characterized by the presence of a Δk between rear and front edges. The cell stops when Δk = 0 nN/µm (Figure 3d). The rapid increase in the cell spring constant at the rear occurs while the cell is moving during the first 90 min of the experiments, whereas the value at the front is almost constant, showing that cell motion maintains a constant rigidity near to the protuberance.

Moreover, the relation between the local change in the cell rigidity and the regulation of cell height is intriguing. It has been shown that the volume of a cell can be directly changed through application of an external osmotic pressure or under stiffer extracellular environments. This hyperosmotic environment triggers effluxion of water out of the cell, which also decreases cell volume, leading to a significant change in cell mechanics and resulting in an increase in rigidity [19,24]. These effects of environment-induced morphological change suggest that cells can adapt their volume through water effluxion. This raises the question whether cells are able to modulate their mechanics and behavior in unchanging environment. Volume decrease under pressured or confined environment can be interpreted as a passive response to a change in external condition. In our case, we suggest an active process where volume change is due to water expulsion from the cell, but as a consequence of a spatially and temporally controlled increase in the cortex rigidity, at the rear and thereafter at the middle of the cell. It has already been shown that cells can migrate in an amoebic mode through plasma membrane flow toward cell rear. This moves cells by exerting tangential forces on the surrounding fluid. Active rearward surface flow could therefore provide a mechanism for cells to swim forward without adhesion [25,26].

Recently, it was shown that tumor cells confined in a narrow tube are able to modify the spatial distribution of ionic pumps and aquaporins in the cell membrane, which creates a net inflow of water and ions at the leading edge and a net outflow of water and ions at the rear, leading to cell locomotion [27,28,29,30,31]. These cells can migrate in the tube even when actin polymerization is suppressed, whereas the inhibition of flow exchange stops the motion. Therefore, cell-volume regulation via water permeation is an alternate mechanism of cell migration in confinement. As confinement affects membrane rigidity, we suggest that this behavior can be part of the regulation of healthy cells motion through the polarization of cell spring constant. Interestingly, this mechanism of rearward surface flow has been found to drive the motion of supra cellular clusters [32].

AFM measurements were repeated on several cells in order to confirm our analysis on the relation between cell mechanical properties and motion (See Appendix A). As shown previously, cells motion is characterized by rear-front Δk gradient in the direction of migration. Δk decreases linearly when cells move in one direction. It is clearly the case for single isolated cells (Appendix A). When cells are in contact to other cells or linked through junction or protuberance, we observe that the mechanism of cell migration depends strongly on the interaction between cells. This induces changes in the direction of motion or rotation, characterized by a more complex evolution of Δk according to cytoskeleton reorganization (Appendix A). We extracted and plotted the evolution of Δk during the linear motion period of the cells (Figure 4a). Fittings show similar values in the spring constant dissymmetry between rear and front edge of the cells, with a mean value to 21.9 ± 3.1 pN/μm^2^ (Figure 4c). This result confirms the dependence between spring constant gradient and cells motion. We also observed that cellular movement generally involves drastic changes in shape which are driven by the cytoskeleton. Cells height change was found to be a relevant indicator of this evolution (Figure 4b). During linear motion, cells show generally a significant decrease in height, to the order of 77 ± 11 nm per μm traveled (Figure 4b, blue, purple, sky-blue, and black curves). For a rounded cell (Figure 4b, green curve), the decrease in height is larger to 270 ± 20 nm/μm, whereas for a spread cell (Figure 4b, red cure), the height increases slightly, to 20 ± 4 nm/μm when the cell detaches from neighboring cells and releases the membrane tension (Figure 4c). Indeed, the evolution of the cells height traduces the effect cytoskeleton activity and also network breaking during motion. Overall, from a mechanical point of view, cells motion is generated by a gradient in rear-front spring constant that decreases during motion and leads to a decrease in cell height. This behavior is generally reversed while cells stop or rotate.

### 2.5. Spatial Polarity and Related k-Dissymmetry

To complete the analysis of the mechanism that leads cell motion, we examined the effect of spatial polarity defined by k-dissymmetry onto the motion of the rear and front edge of the cell described in Figure 1, Figure 2 and Figure 3. For each topographical image, we measured the height profile along the cell axis (Figure 5a, black arrow) and plotted them according to their respective position (X, Y) reported in Figure 2a. The Figure 5b shows some of the profiles measured at different times of the cell displacement. It shows that, as well as the cell height (Figure 2b), the cell shape changes because the front and the rear edges do not behave equally during cell motion. At the start of the experiment, the rear displacement is larger than the front (Figure 5b, profiles #1 and #6). This behavior, concomitantly with volume decreases and cortex rigidity increases, matches with our hypothesis that the cell is expulsing water from the rear. Then the front edge moves suddenly as the rear stays unmoved (Figure 4b, profiles #9 and #10) and the cell height increases drastically (Figure 2b, red arrows). We suggest that this jump originates from the partial detachment and extension of the protuberance at the front edge when the cell migrates. The difference in shape for the profiles #1 and #15 illustrates the changes in the distance between the rear and the front edges of the cell. Overall, the motion is characterized by crushing of the rear edge of the cell while the front stay unmoved until it jumps forward along the motion axis.

For each of the 40 spring constant images, we calculated the position along the cell axis of the rear, middle, and front part of the cell (See Section 3). Then we plotted the relative expansion ε as the length between rear, middle, and front edge respectively (Figure 5c). We also plotted the cell dilatation as the position of rear and front edge with respect to the middle of the cell (Figure 5d) to characterize the shape of the cell during motion. This analysis permits recounting the different phases of morphological changes used by the cell to displace. From the start of the experiment and during the phase where the cell is moving faster (0 min< t < 60 min), ε_front-rear_ is negative. That means that rear motion is larger, or faster in dynamic consideration, than front. It corresponds to a mechanism of rear retraction followed by a front motion when ε_front-rear_ becomes positive after front jump at t = 60 min. Thereafter, the front retracts; this corresponds to the moment when the cell meets another cell in its trajectory. The cell rotates and builds a new protuberance that extends at t = 90 min, the rear spreads as well at t = 120 min. The magnitude of dilatation represents 50% of the cell length at this moment, and this illustrates the large extensibility of the cell. The cell stops at t = 130 min as expected from the large spreading observed previously. During stop, the cell contracts again (also its height increases) before the rear extends again (as well as its rigidity decreases again) at the end of the measurements. Probably, the cell was moving when we started the measurements. We first observed a phase where rear retracts, and where rigidity is equilibrated by expulsing water, and we suppose it has a role in the dynamic of cell anchoring at the front, and cell junction formation in general. We did not capture the mechanism that ensures the continuity of the motion. The cell met an obstacle and changed its trajectory. This process destabilized it and as a consequence it spread and stopped. But it allowed us to observe how a cell tries to adapt to the surrounding during its motion.

The motion of eukaryotic CRFK cells shows several phases characterized by uncorrelated changes in the positions of rear and front parts of the cell. The extensibility of the cells was also characterized by large variation in its height. The motion is therefore an assembly of several morphological changes where k-polarization seems to act as a trigger. At least, the mechanical energy stored in the cells through the rigidity gradient Δk may play a significant role in the process of cell body translocation and rear retraction during displacement. The evolution of k as well as volume increment, are complexes to explain precisely when cells are not moving linearly. The concomitance of several phenomena that generate the change in cells direction or rotation along a new junction induces distinct evolutions of the motion or spreading of the different parts of the cells. In 3D-Environnement context, such as living tissues, this mechanism would guide efficiently on its trajectory as extracellular matrix (ECM) surrounding cell pressure would help rear retraction and push the cell in its direction of motion. Cell migration is hypothesized to involve a cycle of behaviors beginning with leading edge extension. However, recent evidence suggests that the leading edge may be dispensable for migration [33,34], raising the question of what actually controls cell directionality. Thus, the gradient of rigidity may be a part of the cell polarization mechanism necessary to establish and maintain the direction of cell migration. Moreover, we may suppose that active rear retraction prevents from large extension that could damage the membrane when the front edge is extending.

It is probable that the gradient in k must be reconstructed regularly as well as the height (volume) of cells must be regenerated in a sort of breathing mechanism. This work is therefore only a partial description of the mechanical properties of cellular migration. It is also unclear whether these phenomena are essential in the migration process or a secondary mechanism that cells use under certain conditions. But the fact that k-dissymmetry is always observed in the measured moving cells is a good indicator that it play an important role in the CRFK cells migration. A statistical treatment of a larger amount of cells data may probably help to understand to what extend it contributes to the variety of motion mode already described for eukaryotic cells. We anticipate that the motion mechanism observed with CRFK cell may be transposed to a plurality of other eukaryotic cells, given the common nature of their structure and function.

Although we have demonstrated that it is possible to perform AFM studies in living cells over long periods at high temporal and spatial resolution, a clear identification of cytoskeletal function and membrane permeation using inhibitors drugs during imaging may be one pathway toward future discoveries in this area. Moreover, complementary techniques such as advanced light microscopy and spectroscopy in conjunction with the AFM technique presented in this article will have a major impact to elucidate the sophisticated function of the cell machinery.

## 3. Materials and Methods

### 3.1. Cells Preparation

CRFK cells (ATCC CCL-94) were grown in Dubelcco’s modified Eagle’s medium (DMEM; high glucose with glutamine and sodium pyruvate, Gibco (New York, NY, USA)) supplemented with 10% fetal bovine serum (Gibco), 1% MEM NEAA (100×), 2% antibiotic antimycotic (100 mL^−1^) (Sigma (St. Louis, MO, USA) solution, and incubated in a T25 flask at 37 °C in humidified atmosphere with a 5% CO_2_ incubator. Cells at 80% confluency were detached using 0.2% trypsin and 0.02% EDTA in PBS for ~120 s. Trypsin/EDTA solution was replaced with culture medium and 5% Vol. was seeded into a 19 mm diameter round glass substrate coverslips inside a 35 mm diameter round Petri dish for 24 h under the same conditions to reach 50% confluency before measurement.

### 3.2. Optical Microscope

Previously, same CRFK cells were detached as described before and seeded into a support (µ-Slide 8 Well ibiTreat) for 24 h to reach 50% confluency before measurement.

The support with the cells was placed in an incubation chamber with a controlled atmosphere (5% CO_2_ and 37 °C) mounted on an inverted microscope and then allowed to stabilize for 20 min. 275 Time-lapse images were taken for 147 min of five different representative fields with an invert microscope (Zeiss Axio Observer, Jena, Germany) with a confocal unit (LSM 888, Objective: Plan apochromatic 20×/0.8 M27. Beam splitter: MBS: 488. Laser 488 nm; 0.2%). The images were processed and the cells were tracked with ImageJ2 software.

### 3.3. AFM

CRFK is an important cell line for the study of growth and purification of certain viruses and vaccine viruses. For a clear understanding of viral infection process with AFM, it is compulsory to investigate the mechanical properties of these cells during life cycle pathway including migration. The glass substrate with cells was placed in a homemade liquid chamber that maintained the cells at 36 ± 1 °C during the experiment. Classical medium was replaced by transparent medium (complete transparent DMEM+ antibiotic+ 10% fetal bovine serum) with 5 mM HEPES to maintain the pH as the sample cannot be exposed to 5% CO_2_ gas during AFM measurements. Nevertheless, some AFM systems with commercial chamber can supply CO_2_ and HEPES is not always required.

The sample was repeatedly imaged with a commercial Nanotec Cervantes AFM (Nanotec, Madrid, Spain) in jumping mode. We used RC800PSA Olympus silicon nitride cantilevers (Olympus, Tokyo, Japan) with a 0.05 N/m nominal force constant, <20 nm tip radius (15 nm typical), and 18 kHz nominal resonance frequency. The spring constant for the cantilever was calibrated before each experiment by Sader’s method [35]. The settings used were the following on CRFK cells: Imaging 128 × 128 points, applied force: 1.3 nN; speed: 0.428 line/s. The elapsed time during the experiments is calculated from the time it takes to perform a single image. Prior to imaging, the applied force was first calibrated by single force vs. Z piezo displacement curves (FZ) on the glass surface next to the cell.

High resolution images were obtained with jumping mode AFM (JM-AFM), in which the force between the tip of the cantilever and the probe is kept constant and controlled using an electronic feedback loop after each interaction point during imaging. The main inconvenience of jumping mode is the fact that the withdraw-approach procedure of the piezo element must be repeated at every measuring point, this prolongs the scanning time as you need to minimize hydrodynamics noise during approach and it takes long time to perform one single image. As an analysis of the force distance curve was performed after each imaging point, the jumping mode permits the mapping of the spring constant and adhesion of the sample at the same time as topographic imaging.

Usually, cell rigidity analysis provides a measurement of membrane Young modulus by using Hertz-based model such as Sneddon model with a single point applied force and indentation depth. Here the feedback control returns the slope α of the force-distance curve that may be analyzed in a selected segment of the curve from contact point to the maximum applied force. We could calculate the indentation depth assuming the nominal applied force to be constant, but this should introduce an additional error. Moreover, in these cases of large indentation, the classical Hertz/Sneddon models do not describe the difference in the stiffness of the internal structure [15].

We choose to make use of the untreated α values and model the cell and cantilever spring constants by two springs in series. For the determination of the slope of the force distance curves, the software allows setting the first point as well as number of points used to fit a line to the contact region of the force distance curve. We used 124 points for the whole curve from the jump to contact, avoided the first 5 points from the maximum force, and selected the following 30 points to determine the value of the slope where the force-indentation curve is linear (Figure 3b, inset). R square value varies between 0.97 and 0.99 using these parameters. The spring constant for the cantilever was determined by calibration on the substrate. From the slope of the FZ curves, the spring constant of the cell membrane (k_cell_) was calculated as k_cell_ = k_c_k_eff_(k_c_  −  k_eff_)^−1^, where k_c_ is the cantilever spring constant and k_eff_ is the effective spring constant due to cantilever bending and cell membrane deformation.

All the measurements and following image analysis were performed with the help of the WsXM program [36]. Each image was individually treated using local plan fit and “flatten plus” that provide filter to eliminate slopes, low frequency noise, and shadowing effects. Then the sequence of frames was subsequently displayed as a movie. In order to avoid thermal drift of the device, the frames were aligned using a tracking method that compares the first image of the movie with subsequent images by calculating the cross correlation between them. The topographical parameters of the cell, height, volume, and area were calculated using the flooding method of the software. The position of the cell was calculated by measuring the coordinates of the maximum cell height, taking as reference the left bottom corner of the aligned images of the movie. This requires setting of the images values above and below a threshold by the user. The calculation is then performed by the software on the flooded region. For spring constant calculation in Figure 3c, for each image we calculated the average value ± RMS of k_cell_ on a 5 × 5 pixels square around the selected point at the rear, medium and front of the cell-axis respectively. For cell shape calculation, we used the profile method of the software for each image separately and aligned each profile on the same axis according to their respective coordinate calculated in Figure 2a. Then each cell profile was normalized, and the position of the rear and front were calculated as the x-value where the cell height is 0.8 (80% of the maximum cell height) from left and right of the cell top respectively. Middle is taken as the point on the cell axis where cell height is maxima. Data analysis and representation was performed with Origin 8.5 (www.origin.com, accessed on 21 January 2015).

## 4. Conclusions

In summary, we have investigated the mechanical properties of eukaryotic CRFK cells during motion on glass substrate under culture medium at 37 °C. We showed that sequential imaging with AFM brings information on the evolution of cells morphology and mechanical properties during different displacement phases.

Migration is accompanied by changes in shape and volume that are thought to be caused by the polarization of the cellular spring constant along the motion axis. This gradient in rigidity explains the rapid decrease in the cell height and the retraction of its rear part. The rear of cells is then mechanically active during motion, and this has important implications in contraction-extension mechanisms of rear and front edges. Our study represents a step toward generating a detailed map of mechanical properties correlated to the cytoskeletal organization and to the migratory state of eukaryotic cells, which is necessary for the comprehensive understanding of the biophysics of their motility. Furthermore, it highlights the potential of AFM for the understanding of cellular biophysics through sequential real-time imaging of mechanical properties in physiological conditions.

## Figures and Tables

**Figure 1 ijms-23-14369-f001:**
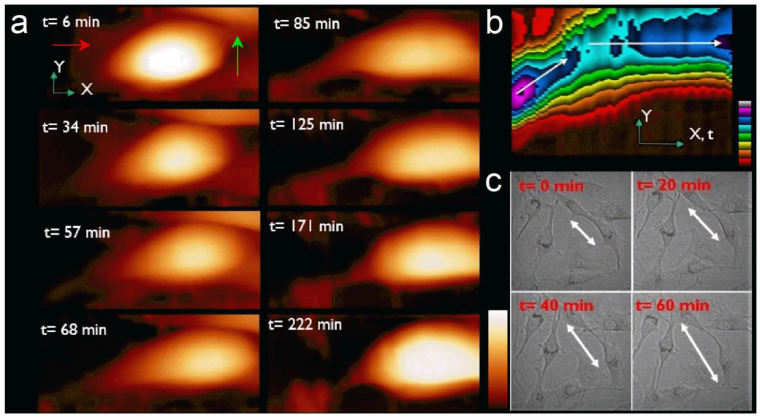
(**a**) Series of topographic images (70 µm × 31 µm) recorded in real time for a single CRFK cell during its motion on glass surface for t = 6, 34, 57, 68, 85, 125, 171, and 222 min respectively. Z-scale = 3.5 µm. Some residues (red arrow, t = 6 min) and protuberance (green arrow, t = 6 min) are visible from the start of the experiment. (**b**) Kymograph of the trajectory of the cell with time along the X–Y defined from the borders of the images. The cell starts moving about 45° from X-axis during 90 min. Then it changes direction to follow X-axis during the rest of the experiment (Z-scale, colors, 4 µm). (**c**) Optical microscope images (200 µm × 200 µm) of the motion of CRFK cell onto glass substrates show the dynamics and morphologies of the cells anchored to each other during movement (white arrow).

**Figure 2 ijms-23-14369-f002:**
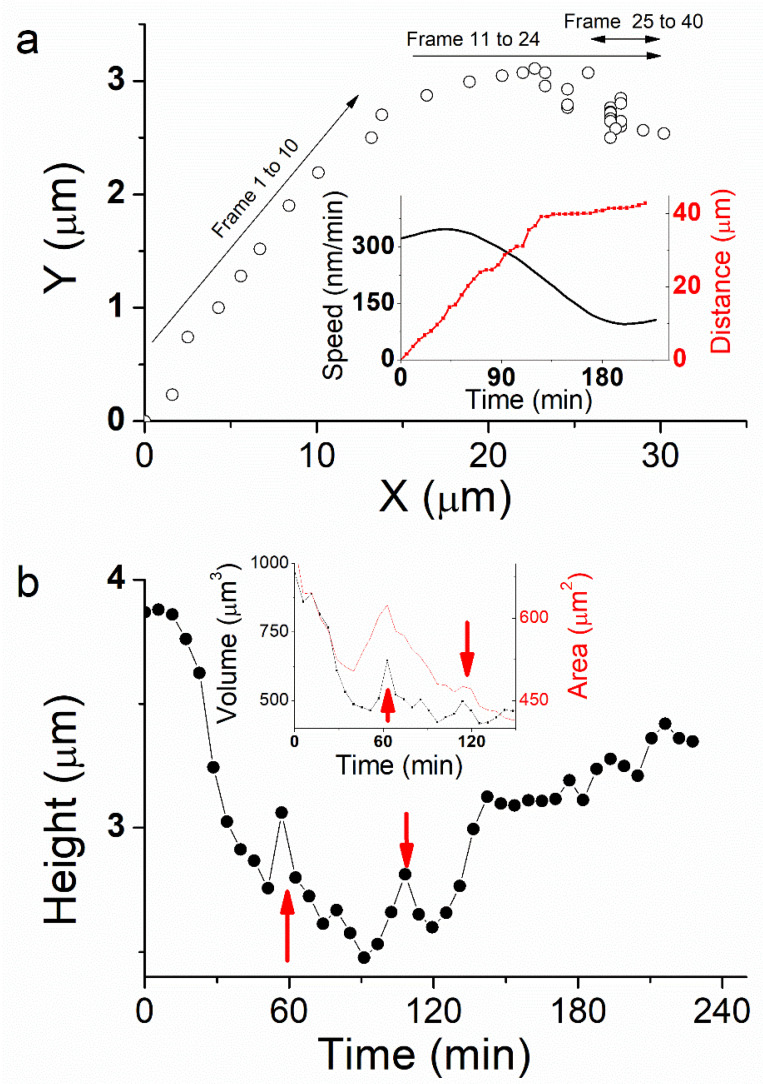
(**a**) Trajectory of the cell plotted as the position of the cell for the aligned images. Each point (open circle) corresponds to the position of the cell in a single frame. Cell speed and total traveled distance are calculated from the plot (Inset black and red curves respectively). (**b**) Cell maximum height, as well as volume and area (Inset black and red curve respectively) is calculated from each image during cell motion.

**Figure 3 ijms-23-14369-f003:**
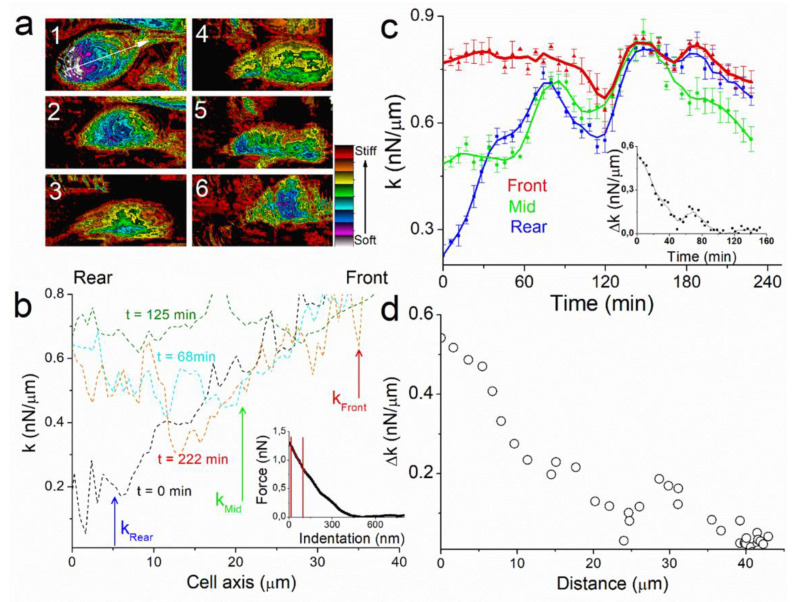
(**a**) Spring constant images of the cell (70 µm × 31 µm) at t = 0, 57, 85, 105, 130, 217 min respectively that show the gradient in k along the cell motion axis (Image 1, white arrow). Z-scale: 1 nN/µm. The spatial resolution used for each image was 128 × 128 pixels, this corresponds to 16,384 indentations performed for each map during cell stiffness characterization. (**b**) Profile of the k-image along the cell motion axis for t = 0, 68, 125, 222 min, respectively. The blue (rear), green (mid), and red (front) arrows indicate the position where k was calculated. The inset represents a force–distance curve performed onto the CRFK cell with the region (red lines) where the spring constant k is calculated. (**c**) Real time plot of the spring constant k measured for three distinct points of the cell along the motion axis at the rear (blue), middle (green), and front (red) respectively, with the evolution of Δk_rear-front_ (inset). (**d**) Plot of Δk_rear-front_ as function of the cell’s traveled distance.

**Figure 4 ijms-23-14369-f004:**
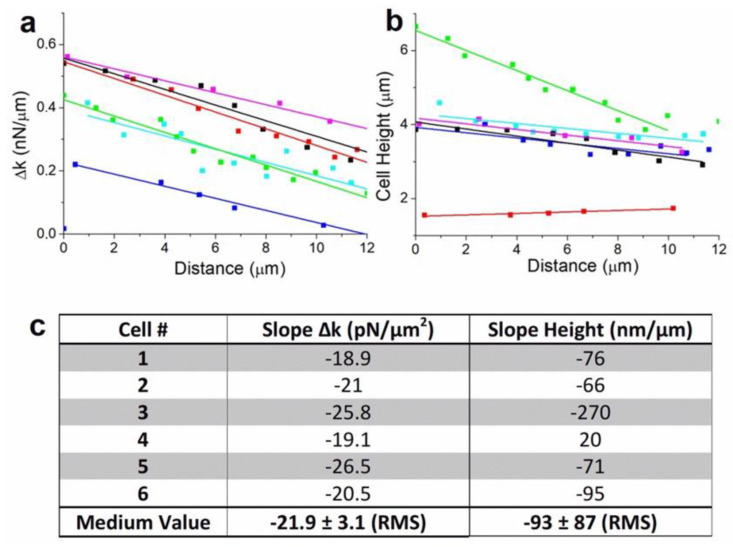
Analysis of the results for six different cells: Plot of (**a**) Δk_rear-front_ and (**b**) cells height, as function of the traveled distance with linear fitting of the respective curves. (**c**) Calculated values of slope of (Δk) and cells height for the six experiments.

**Figure 5 ijms-23-14369-f005:**
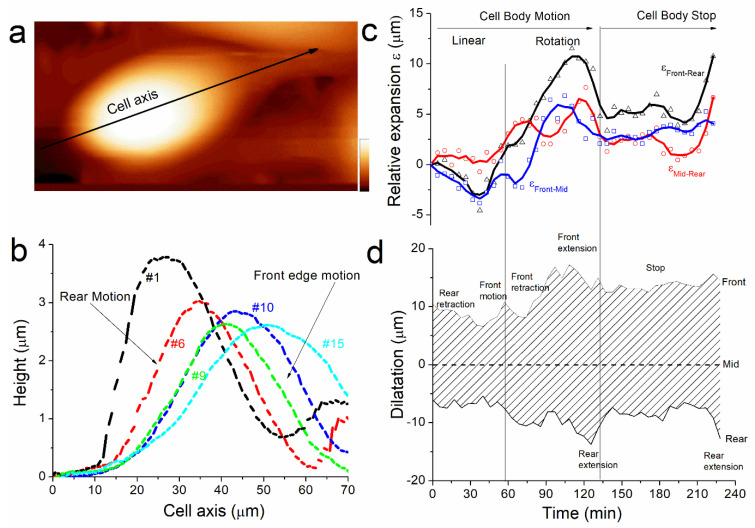
(**a**) Topographical image of the cell (70 µm × 31 µm) at t = 0 min that shows the cell axis where the profile is measured (black arrow). Z-scale = 3.5 µm. (**b**) Profiles of the topographical images along the cell axis with respect to their traveled distance for t = 6, 34, 51, 57, 86 min respectively (images 1, 6, 9, 10 and 15 respectively). Real time plot of the, (**c**) relative expansion, and (**d**) dilatation, during cell motion that shows the different phases of rear retraction and front extension, rotation, and stop of the cell.

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
