# Peer review of "Eukaryotic CRFK Cells Motion Characterized with Atomic Force Microscopy"

_ijms, 2022, doi:10.3390/ijms232214369_

Round 1

Reviewer 1 Report

Although the subject of the study is interesting, the methodology and instrumentations are at top level, the results seem to be purely descriptive. The high resolution stiffness maps are absolutely great achievements, but only one step further is to use it to compare the observations to transgenic or at least to some perturbed cellular states (membrane, cytoskeletal or other adhesion or elasticity related modifications). 

Notes and suggestions:

How much indentation was suffered the cells during mapping? Could the authors provide an indentation map at least in the supplementary section, in order to appreciate what exactly gives the stiffness map result? Few hundreds of nanometers might be only cortical, but larger would be influenced by cell nucleus at central region.

The author do not discuss how universal is their finding, here two questions are to be discussed: one is for cell type, namely the presented representative maps and features how often occur in this way (maybe some statistics would be benefical as well), the second is some comparison for other cell types (this is slightly addressed in Discussion session but some more elaboration would be good).

The section of Results, contains some citations and parts which are discussing the results. Either rename the section Results and Discussion and merge with next "Discussion" section or remove the sentences with references and place them to Discussion.

It would ease the understanding of the results presented if the color bars for elasticity maps would have some labels, to instantaneous identification of which color codes for stiff and soft regions. In some cases even figure caption lacks to provide the z-scale range of color code Fig5a. Supplementary figures does not have color bars at all, hence it is impossible to identify which color codes for hard or soft regions and how large differences occur.

Although the Fig4 provides nice data for the 6 cells presented, the low number of experiments (cells monitored) is a bit blurring the brightness of the findings.

Discussion section appears in majority suitable for introduction rather than real discussing and comparing the results. Please rewrite.

I would support the publication of the manuscript after all issues were substantially improved.

Reviewer 2 Report

Dear editor,

I have read with interest the work by Zamora-Ceballos and coworkers in which they detail an AFM study of the dynamic properties of wukaryotic CRFK cells during their motion.

The manuscript is clear and very interesting, and the conclusions detail with good vision some subsequent experiments that can enlarge the scope of this field of research.

I have only few comments that should be addressed before this manuscript can be published:

The authors should reread the manuscript to correct the few English errors and misprints.

Figure 1c is discussed in the Results section after the discussion of Figure 2. This is somewhat confusing.

Do the optical images in Figure 1c correlate directly to the AFM images in Figure 1a?

In the discussion of Figure 2a, the authors refer to "frames" which are not detailed in the figure or in the caption (Line 97 and Line 99, for instance).

How are the frames correlated to the elapsed measuring time?
